# Epidemiology, Microbiology and Severity of Bronchiolitis in the First Post-Lockdown Cold Season in Three Different Geographical Areas in Italy: A Prospective, Observational Study

**DOI:** 10.3390/children9040491

**Published:** 2022-04-01

**Authors:** Anna Camporesi, Rosa Morello, Valentina Ferro, Luca Pierantoni, Alessandro Rocca, Marcello Lanari, Gian Luca Trobia, Tiziana Sciacca, Agata Giuseppina Bellinvia, Alessandra De Ferrari, Piero Valentini, Damian Roland, Danilo Buonsenso

**Affiliations:** 1Anesthesia and Intensive Care Unit, “Vittore Buzzi” Children’s Hospital, 20154 Milan, Italy; anna.camporesi@gmail.com (A.C.); alessandra.deferrari@asst-fbf-sacco.it (A.D.F.); 2Department of Woman and Child Health, Fondazione Policlinico Universitario A. Gemelli IRCCS, 00168 Rome, Italy; rosa.morello91@gmail.com (R.M.); piero.valentini@unicatt.it (P.V.); 3Department of Pediatric Emergency, Bambin Gesù Children’s Hospital IRCCS, 00168 Rome, Italy; valentina.ferro@opbg.net; 4Pediatric Emergency Unit, IRCCS Azienda Ospedaliero-Universitaria di Bologna, 40138 Bologna, Italy; lucapierantoni82@gmail.com (L.P.); alessandro.rocca4@unibo.it (A.R.); marcello.lanari@unibo.it (M.L.); 5Pediatric and Pediatric Emergency Room Unit, Cannizzaro Emergency Hospital-Catania, 95126 Catania, Italy; trobia@tin.it (G.L.T.); tiziana.sciacca@icloud.com (T.S.); aga.bellinvia@gmail.com (A.G.B.); 6Paediatric Emergency Medicine Leicester Academic (PEMLA) Group, Leicester Hospital, Leicester LE1 5WW, UK; dr98@leicester.ac.uk; 7Social science APPlied to Healthcare Improvement REsearch, SAPPHIRE Group, Health Sciences, Leicester University, Leicester LE1 7RH, UK; 8Center for Global Health Research Studies, Università Cattolica del Sacro Cuore, 00168 Rome, Italy

**Keywords:** bronchiolitis, COVID-19, RSV, respiratory syncytial virus, children

## Abstract

The aim of this study was to understand the epidemiology, disease severity, and microbiology of bronchiolitis in Italy during the 2021–2022 cold season, outside of lockdowns. Before COVID-19, the usual bronchiolitis season in Italy would begin in November and end in April, peaking in February. We performed a prospective observational study in four referral pediatric centers located in different geographical areas in Italy (two in the north, one in the center and one in the south). From 1 July 2021 to 31 January 2022, we collected all new clinical diagnoses of bronchiolitis in children younger than two years of age recording demographic, clinical and microbiological data. A total of 657 children with a clinical diagnosis of bronchiolitis were enrolled; 56% children were admitted and 5.9% required PICU admission. The first cases were detected during the summer, peaking in November 2021 and declining into December 2021 with only a few cases detected in January 2022. RSV was the commonest etiological agent, while SARS-CoV-2 was rarely detected and only since the end of December 2021. Disease severity was similar in children with RSV vs. non-RSV bronchiolitis, and in those with a single infectious agent detected compared with children with co-infections. The 2021–2022 bronchiolitis season in Italy started and peaked earlier than the usual pre-pandemic seasons, but had a shorter duration. Importantly, the current bronchiolitis season was not more severe when data were compared with Italian published data, and SARS-CoV-2 was rarely a cause of bronchiolitis in children younger than 24 months of age.

## 1. Introduction

Since the first description of SARS-CoV-2 in China in December 2019, the COVID-19 pandemic has had a profound impact across the globe. SARS-CoV-2 has caused millions of infections and thousands of deaths worldwide [1] However, the virus has also had other indirect effects on society. Restrictive measures, particularly lockdowns, aimed at preventing the spread of the virus have negatively impacted the economies of almost all countries worldwide [2]. Similarly, school closures, enforced in most countries aiming at reducing local community transmission of the virus, have negatively impacted the wellbeing and social and cultural development of millions of children worldwide [3]. On the other hand, other non-pharmacological interventions widely established, such as social distancing, hand washing, and masking, have also been linked to changes in the epidemiology of other infectious diseases globally [4].

Specifically, a drastic change in respiratory infectious diseases has been reported [5]. During the first pandemic cold seasons in Europe [6,7], United States [8], Latin America [9], and Oceania [10,11,12], a dramatic decline in the influenza and bronchiolitis cases have been described in several studies. Also, pediatric emergency departments have seen significant reductions in the number of febrile children [13]. It has been speculated that more attention to hygiene, routine use of masking, and physical distance played a role in the changed epidemiology of common respiratory pathogens. These changes in respiratory infectious diseases have been particularly evident during the year of the pandemic (January–April 2020), when almost all countries established either full or partial lockdowns, longer periods of school closures, and supported smart working [14]. However, given the negative impact of strict restrictive measures on economics, with the introduction of effective vaccines early in 2021, many countries have chosen less and less to rely on closures, and instead to support massive vaccination campaigns, use of masks in closed environments, and common hygiene measures. As a consequence, the second year of the pandemic has been characterized by more social interactions compared with the first one. Therefore, a theoretical return of common infectious pathogens was expected.

The impact of this was initially registered in Australia and New Zealand, countries that had a successful containment of SARS-CoV-2 and quickly reopened the community to normal pre-pandemic life. In these countries, an unexpected aseasonal surge of bronchiolitis with a high number of cases compared to pre-pandemic periods has been registered [15,16,17,18]. These findings have worried researchers worldwide, concerned that the re-introduction of respiratory viruses within pediatric communities that have never encountered common viruses for about two years might have had a huge clinical impact in terms of number and severity of respiratory diseases. This concept has been called an “immune debt” [19], referring to the theory that the reintroduction of a virus in a community of millions of people that have never met that agent, and therefore has no immune memory toward it, would have led to uncontrolled community transmission and, possibly, to larger number of children with severe disease. Given that acute bronchiolitis is the commonest cause of hospital admission and mortality in young children during cold seasons [20], the concern of most pediatricians was other countries out of Australia and New Zealand might experience a particularly severe bronchiolitis [21], with a possible beginning of the epidemic in unusual seasons compared to pre-pandemic periods.

For these reasons, starting on July 2021 (Italian Summer), we implemented a team of Italian pediatricians working in different geographical areas to prospectively collect all cases of bronchiolitis, aiming to characterize the epidemiology, disease severity, and microbiology of bronchiolitis in the first post-lockdown cold season in Italy.

## 2. Materials and Methods

This is a national multi-center prospective observational cohort study performed in four Italian University Hospitals in three different geographic areas (Northern Italy—Milano and Bologna, Central Italy—Rome, and Southern Italy—Catania) from 1 July, 2022 to 31 January, 2022. The study was adapted by an open access BronchSTART protocol developed by the PERUKI network, aiming to monitor realtime new bronchiolitis cases in the United Kingdom and Ireland since July 2021 (available at https://pubmed.ncbi.nlm.nih.gov/34458589/, last accessed on 30 July 2021).

During the study period, no lockdowns were established in Italy, and masking was mandatory in closed places or in school for children older than five years of age. The study was approved by the Ethics Committee of the participating centers.

### 2.1. Inclusion Criteria

For the purpose of this study, we included all consecutive children assessed in the participating institutions and receiving a clinical diagnosis of bronchiolitis (e.g., children with cough, tachypnoea or chest recession, and wheeze or crackles on chest auscultation) or a first episode of acute viral wheeze [22,23,24]. We included children younger than two years of age for two reasons. First, there is no full agreement on the upper age limit for diagnosing bronchiolitis according to different international guidelines, which varies between 12 and 24 months [22,25,26], despite growing evidence suggesting that the upper age limit might be restricted to 6 months [27]. Secondly, given the hypothesis that children in their second year of life might have missed being in contact with common respiratory viruses linked to bronchiolitis due to restrictive measures during their first year of life, we decided to include also children aged 12–24 months. The hypothesis was that this age group might have suffered a first episode of bronchiolitis or viral wheezing. 

### 2.2. Exclusion Criteria

We excluded children older than two years of age, those with previous episodes of wheeze or those who had previously received a diagnosis of bronchiolitis in the past, which might indicate the diagnosis of recurrent wheeze rather than bronchiolitis.

### 2.3. Outcomes

#### 2.3.1. Primary Outcome

The objective of this paper was to characterize the epidemiological trend of bronchiolitis in children under two years of age presenting to four Italian hospitals located in different geographical areas in the first post-lockdown cold season in Italy. Before the beginning of the COVID-19 pandemic, the distribution of the usual “RSV seasons’’ (which overlap with the bronchiolitis seasons) in Italy ranged from mid-November until the end of April, peaking in mid-February [28].

#### 2.3.2. Secondary Outcome

Secondary outcomes included the following:-To define overall disease severity (defined in terms of need of intensive care unit or different type of ventilation support) in children with bronchiolitis during the first cold season where lockdowns (full or partial) were not implemented in Italy;-To define the main etiological agents in this cohort of children with bronchiolitis;-To compare disease severity in children with RSV versus non-RSV bronchiolitis;-To understand the impact of SARS-CoV-2 on a large cohort of children with bronchiolitis;-To compare disease severity in children with bronchiolitis due to a single etiology versus those with multiple viruses detected; specifically, we defined as “co-infection group” those children that had more than one virus detected simultaneously at the nasopharyngeal swab;-To compare disease severity and main microbiological agents in children with bronchiolitis younger or older than 12 months of age.-To compare the temporal distribution of bronchiolitis cases during the 2021/22 season with pre-pandemic seasons (data available only for Bologna, Rome, and Catania).

### 2.4. Data Collection

Anonymous data have been collected on an online dataset shared by the PIs of the four participating centers. Given the nature of the study and the peculiar period characterized by high workload in hospital settings and limited resources available, along with the need of having real time prospectively collected clinical data, each center was allowed to establish its own guidelines on the viral testing and therapeutic management of children (since type of treatment was outside of the purpose of the study).

Data were collected at time of the first hospital evaluation of the patient. Seven days after this evaluation, families were recontacted (if discharged, or outcome reassessed if hospitalized) to understand if the highest level of respiratory support needed or area of admission have changed during those days (therefore, families have been asked whether during those seven days the child has spent any time within a pediatric intensive care unit (PICU).

At baseline, we collected data about patient demographics, date of initial assessment, main clinical parameters, acuity, treatments administered, admission or discharge, including type of admission ward, need and type of ventilation support, number of siblings, aetiologies of the bronchiolitis (through nasopharyngeal swabs, including SARS-CoV-2), when tests were performed. For non-SARS-CoV-2 etiology, we use the ePlex Respiratory Panel which detects PCR of RSV, influenza, rhinovirus, adenovirus, parainfluenza, metapneumovirus, bocavirus, and enterovirus. In order to facilitate the participating centers, and due to the fact that the epidemiological characterization of the bronchiolitis season was the primary outcome, the performance of the nasopharyngeal swab for respiratory viruses was carried out at the discretion of the attendant physicians. All evaluated children with bronchiolitis underwent SARS-CoV-2 rapid antigen test in the emergency department, while the PCR was carried out for all admitted patients or in case of a positive rapid test.

At seven days, data included the child’s ultimate outcome (alive, death), need of PICU admission, and the highest level of respiratory support needed within the seven days from initial diagnosis of bronchiolitis (low-flow oxygen, high flow oxygen on nasal cannulae (HFNC), continuous positive airway pressure (CPAP), invasive ventilation).

### 2.5. Statistical Analyses

Statistical analysis was performed using the software STATA/IC 14.2 version 2017. We tested the normality by Skewness/Kurtosis test. Data were reported as median values with an interquartile range (IQR), and direct comparisons were made with Mann-Whitney rank-sum tests or Kruskal-Wallis test. Percentages were used to describe categorical outcomes, and distributions of categorical data were compared with either a Pearson’s χ^2^ test or a Fisher’s exact test, as appropriate. 

## 3. Results

### 3.1. Study Population

During the study period, 657 children with a clinical diagnosis of bronchiolitis were enrolled. The overall temporal distribution of new diagnoses from July 2021 to 31 January 2021, and according to the main geographical areas are reported in Figure 1. The first cases of bronchiolitis were reported in the middle of the Italian summer seasons (July and August). Cases then increased significantly in October, peaked in November and December, and showed a fast decline during the second half of December with only a few cases reported during January 2022. Appendix A describes the distribution of bronchiolitis cases during the 2021/22 season with pre-pandemic seasons (data available only for Bologna, Rome and Catania), supporting the observation of an anticipated peak and a shorter duration of the bronchiolitis season in the post-pandemic period.

Clinical and demographic characteristics of the study population are described in Table 1. Overall, 368 (56%) children were admitted and 39 (5.9%) required PICU during the admission period. Within the 7 days since initial diagnosis, 2 children (0.3%) needed mechanical ventilation, 43 (6.5%) CPAP, 128 (19.5%) HFNC, and 167 (25.4%) low-flow oxygen. No children died.

### 3.2. Aetiologies of Bronchiolitis and Impact on Disease Severity

A virus was identified in 216 out of 264 tested children (81.8%). RSV was the commonest isolate (Table 2). Other bronchodilator, ipratropium; corticosteroid treatment, any intravenous or oral steroid treatment administered.

Children with RSV were significantly younger (median age 2 months, IQR 1–5, vs. 5 months, IQR 1.69–11; p 0.004) (Table 3).

The temporal distribution of etiologies is reported in Figure 2. SARS-CoV-2 was unfrequently the etiological agent of bronchiolitis and was only detected since the second half of December 2021.

Disease severity according to etiology is reported in Table 3. Overall, need for admission, including PICU admission and respiratory support (HFNC, CPAP or mechanical ventilation), was similar in children with RSV or on-RSV etiology.

Of the eighteen cases of bronchiolitis due to SARS-CoV2, one required PICU admission (5.5%), two required low flow oxygen support (11.1%), two required high flow nasal cannula oxygen (11.1%), two required CPAP (11.1%), and none required mechanical ventilation. 

We examined if children with multiple viruses detected (co-infections group) had different clinical characteristics and disease severity compared with those with a single etiological agent detected (Table 4). Overall, a higher percentage of co-infected children required any type of respiratory support and PICU admission, but differences were statistically not significant.

Table 5 show the distribution of isolated viruses according to age group. RSV was more frequent in infants younger than 12 months, but differences were statistically non-significant (76.1% vs. 65.2%, *p* 0.25), while rhinovirus was more frequently isolated in children older than 12 months (19.7% vs. 43.48%, *p* 0.009). SARS-CoV-2 was only detected in children younger than 12 months of age presenting with bronchiolitis, although the small number of children with SARS-CoV-2 infection did not allow statistical comparisons. 

We analyzed the distribution of main viruses according to the prematurity groups of children with bronchiolitis (Table 6). RSV was the commonest agent in all groups, and overall the distribution of main viruses was similar.

### 3.3. Disease Severity According to Age Groups or Prematurity

The need for respiratory support or PICU admission according to age or prematurity are reported in Table 5 and Table 6, respectively. Overall, children older or younger than 12 months had a similar probability of being admitted in the PICU, or receiving respiratory support (HFNC, CPAP or invasive ventilation) (Table 5). Although children born <34 weeks of gestational age more frequently needed PICU admission, HFNC, CPAP or invasive ventilation, differences were not statistically significant (Table 6).

## 4. Discussion

In this study, we have prospectively collected all children diagnosed with bronchiolitis younger than two years of age in four hospitals covering three different geographical areas in Italy. Overall, we found the bronchiolitis season started earlier than usual and disease severity was not different than expected by available literature. To the best of our knowledge, this is the first European report of an altered bronchiolitis season in the first year of the pandemic managed without lockdowns in Italy.

During the first year and a half of the pandemic, the number of cases of bronchiolitis drastically decreased worldwide [5]. According to some authors, changes in human relationships and the widespread use of masks, higher attention to hygiene measures, school closures and lockdowns were the main factors contributing to this changed epidemiology [5]. However, with new effective COVID-19 vaccines available, the large majority of countries, including Italy, have significantly decreased social restriction measures and reopened schools, therefore the return of bronchiolitis was expected. Interestingly, our prospective collection of clinically diagnosed bronchiolitis in four hospitals allowed us to depict a new trend in bronchiolitis cases. In particular, we found a peak in the diagnosis of bronchiolitis in the post-pandemic 2021–2022 season in October–November 2021, with a progressive reduction of new cases since the second half of December 2021 and only a small number of cases reported in January 2022. Compared with historical Italian published data, the distribution of the usual Italian “RSV seasons’’ (which overlap with the bronchiolitis seasons) ranges from mid-November to the end of April, peaking in mid-February [28], as mostly happens in other European countries [29]. Moreover, from the centers in Rome, Bologna, and Catania we had pre-pandemic data available in Appendix A, confirming that the 2021–2022 season showed an earlier peak compared with 2018–19 and 2019–20 seasons. The early peak we documented in Italy is in line with the earlier, unusual peak documented in Australia [15,16,17,18]. Moreover, a peculiar finding was the evidence of a trend toward an early decline of new cases in the second half of December and was maintained in January 2022. This decline was also confirmed during the writing of this manuscript (almost zero cases in February 2022, data not shown), supporting the documentation of an overall shorter bronchiolitis season in Italy.

Some authors were worried that those children that missed the previous bronchiolitis season during 2020 might “pay the immunological debt” of missing viral infections during the first years of life [19]. Specifically, there was concern that virus circulation among a pediatric population that did not develop any previous immunity toward respiratory viruses might have led to more cases of bronchiolitis and, possibly, more severe disease. However, we found that overall severity was not different from what expected from historical data. Since there are several available clinical bronchiolitis scores whose assessment may be subjective and changing during disease course or according to the evaluating clinician, we focused on the highest level of respiratory support needed within seven days since the first diagnosis. We found that only two children required invasive ventilation (0.3%) and 43 (6.5%) needed CPAP, representing the most objective parameters of disease severity. PICU admissions were also overall low (39 (5.93%)). However areas of admission can significantly change according to local practices (e.g., CPAP or HFNC performed in the ward or PICU). Although we were not able to collect pre-pandemic details from our centers, two of our participating hospitals (Rome—Center Italy and Bologna—Northern Italy), published pre-pandemic data recently, showing that overall the severity of bronchiolitis in 2021–22 is similar to previous periods [30,31]. Moreover, these findings are in line with the recent data from Australia during the second half of 2020, when restrictions were relaxed [18]. The authors found that despite an earlier RSV peak, there was an overall reduction of RSV-coded hospitalizations and PICU admissions (including bronchiolitis-related hospitalization in children younger than 24 months) in 2020 compared with previous pre-pandemic years [18]. Also in Paris, an out-of-season bronchiolitis peak was associated with lower PICU admissions or respiratory support compared with previous epidemics [32]. Our findings, along with the mentioned studies, do not confirm the estimates conducted by other authors that suggested a possible significant increase in RSV hospitalizations after viral reintroduction within the communities [33].

For this study, we fixed the upper age cut-off for a diagnosis of bronchiolitis at 24 months for two reasons. First, there is no international consensus on the age cutoff of bronchiolitis [22,25,26]. More importantly, since most children in their second year of life missed the bronchiolitis seasons during 2020 (which was their first year of life), we speculated that this group of children might have experienced the first episode of bronchiolitis in their second year of life. In this regard, Australia and France both documented a shift toward higher mean ages of children affected by RSV during the out-of-seasons bronchiolitis seasons [18,19]. In our study, we found that up to 1 every 5 children with bronchiolitis were older than 12–14 months old and, importantly, overall severity in terms of PICU admissions and highest respiratory support were similar in the two groups. Interestingly, RSV was significantly more commonly isolated in children younger than 12 months of age.

From a microbiological perspective, RSV was the most common agent, in accordance with what was expected from pre-pandemic data, followed by Rhinovirus [34,35,36,37]. A similar number of children within the RSV and non-RSV group required mechanical ventilation, however, RSV-bronchiolitis required more frequently CPAP or PICU, but differences were statistically non-significant. Since most children have missed the previous bronchiolitis season and might have not developed immunity to most viruses, we have hypothesized that those with co-infections could have had a more severe disease. However, in our analyses of children with bronchiolitis due to a single virus or multiple ones, overall disease severity was very similar.

A relevant finding of our study is that SARS-CoV-2 was rarely isolated in children with bronchiolitis and that the only cases were detected during the second half of December 2021. This finding is peculiar and deserves attention. Although virological PCR data were not available for all, in the four participating centers all children evaluated in the emergency department for respiratory symptoms undergo a rapid SARS-CoV-2 test and all those hospitalized perform a SARS-CoV-2 nasopharyngeal swab. Therefore, the low number of SARS-CoV-2 cases in children with bronchiolitis is realistic and is in line with reports from other countries [38]. Interestingly, the only cases have been detected since the end of December, a period characterized by a steep rise in SARS-CoV-2 community transmission in Italy, in all age groups [39]. This temporal overlap suggests that SARS-CoV-2 is an infrequent cause of clinically significant bronchiolitis in children requiring emergency department evaluation and this may happen more frequently in case of massive community circulation. This finding, if confirmed by larger studies, can inform isolation policies in hospitals during peak of bronchiolitis cases. Interestingly, SARS-CoV-2 caused bronchiolitis only in children younger than 12 months of age and that might suggest similar pathological events happen in the peripheral airways in young infants after viral infections, rather than virus-specific induced mechanisms. Also, none of the COVID-19 cases were severe invasive ventilation but six needed a non-invasive respiratory support (33.3%), similar to what reported by a Spanish study [38]. However, one of the most interesting findings is that bronchiolitis diagnosis dropped as soon as SARS-CoV-2 peaked in Italy, and first SARS-CoV-2 bronchiolitis appeared. Importantly, the trend of very low numbers or no new bronchiolitis persisted during February 2022 (data not shown). This finding is surprising and may have several explanations. First, given the high circulation rate, it is possible that the population’s adherence to preventive measures has been stricter since December 2021, leading to a lower transmission of respiratory viruses. Second, with very high daily cases of SARS-CoV-2 infection, it is possible that thousands of people were in quarantine due to infection or known exposure and, therefore, less people were involved in routine life and less children went to school. This scenario, despite no implementation of formal lockdowns, could indirectly lead to lower circulation of respiratory viruses. Third, we cannot exclude a mechanism of “viral interference”, a phenomenon in which two viruses interact within a host, and the immune responses toward one of them reduce the possibility to replicate of the other [40]. This phenomenon has been recently demonstrated by Wu and colleagues [40]. Infecting upper airway epithelial cells with Rhinoviruses, they documented an increased expression of interferon-stimulated genes that protected against Influenza A virus infection, while the block of these interferon responses restored influenza A virus replication [40]. These in-vitro findings confirm the epidemiological observations that during the 2009 Influenza A epidemic the annual autumn rhinovirus epidemic interrupted and delayed the transmission of the emerging influenza virus [40,41,42,43]. Therefore, we can speculate that SARS-CoV-2 infection in the upper respiratory tract can interfere with the replication of other respiratory viruses and, therefore, in case of high community SARS-CoV-2 transmission, the overall circulation of common respiratory viruses is reduced in both adults and children. To support this hypothesis, in hamsters, SARS-CoV-2 infection significantly reduced influenza virus titers in the nose and lungs [44]. This scenario may suggest that the major reductions in respiratory infections reported during the first waves of the pandemic [4,5] might have been supported by both social behavior (e.g., social restrictions and other preventive measures) and microbiological reasons (interference).

Our study has limitations to address. Although this was a prospective observational study and all consecutive clinically diagnosed bronchiolitis have been included, we were not able to guarantee the performance of nasopharyngeal PCR tests for respiratory viruses to all children for a number of reasons. First, resources differ in the hospitals and this study was not funded. Second, given the epidemiological importance of understanding new trends in bronchiolitis, we could not implement a system able to guarantee testing to all children and decided to focus on a minimal dataset which allowed us to collect all clinical diagnosis and more clinically-relevant outcomes. However, this approach reflects real-world clinical practice and we were still able to collect a high number of microbiological diagnoses. Conversely, SARS-CoV-2 rapid tests are routinely carried out in our Institutions. Although rapid tests can give false negative results, all admitted children undertook SARS-CoV-2 PCR tests. Therefore, although SARS-CoV-2 cases might have been underestimated in discharged children, it is realistic to hypothesize that SARS-CoV-2 accounted for a minimal number of bronchiolitis. A last limitation is the lack of a comparison with pre-pandemic periods. However, since the literature is full of studies on the seasonality and severity of bronchiolitis in children, due to time and resource constraints we decided to focus on the novelty of this season.

## 5. Conclusions

In conclusion, our study showed that the expected 2021–2022 bronchiolitis season in Italy started and peaked earlier than the usual pre-pandemic seasons but had a shorter duration. Also, this season was not more severe than the previous ones. Importantly, SARS-CoV-2 was rarely a cause of bronchiolitis in children younger than 24 months of age. More prospective, multinational data during the following years are needed to understand how the SARS-CoV-2 pandemic affect the routine circulation of common respiratory viruses, which will help in better understanding the epidemiology and interconnection of respiratory viruses and, in turn, can inform better policies to prevent outbreaks or manage future pandemics.

## Figures and Tables

**Figure 1 children-09-00491-f001:**
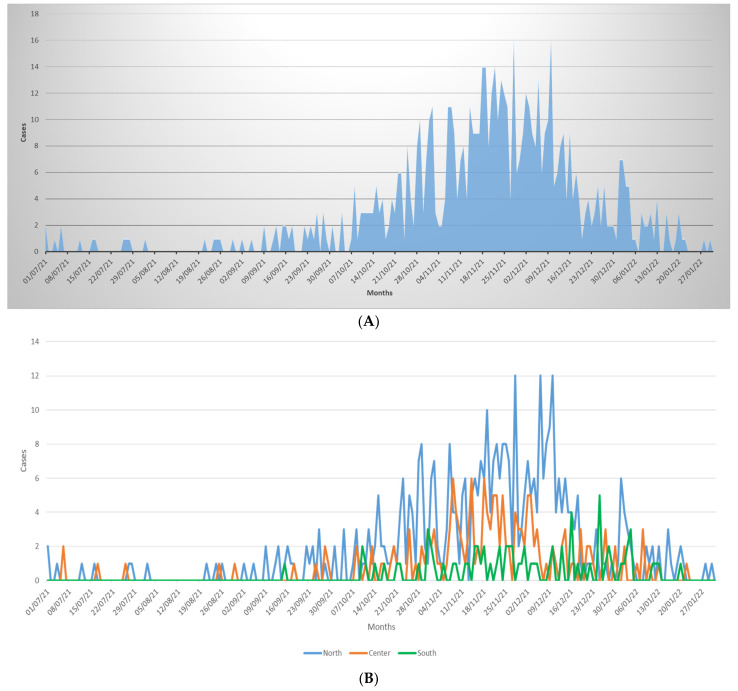
Temporal distribution of cases of bronchiolitis during the study period (**A**) and according to the main geographical areas (**B**).

**Figure 2 children-09-00491-f002:**
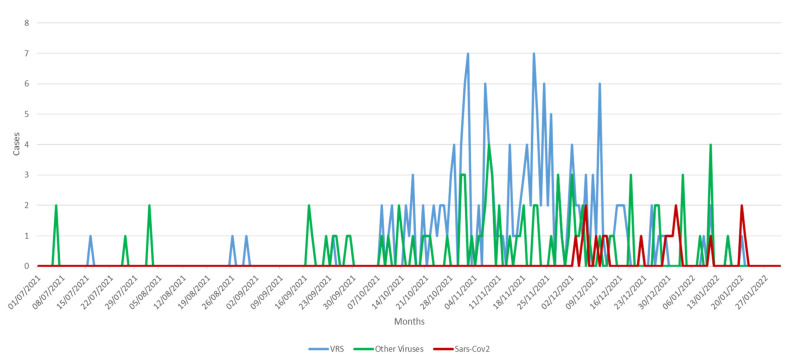
Temporal distribution of cases of bronchiolitis according to the main etiologies.

**Table 1 children-09-00491-t001:** Study population. CPAP, continuous positive airway pressure; PICU, pediatric intensive care unit.

	Study Population	North	Middle	South	*p*-Value
	*n* = 657	*n* = 410	*n* = 170	*n* = 77	
Sex, *n* (%)					
• Female	368 (56.01)	170 (41.46)	77 (45.29)	42 (54.55)	0.1
• Male	289 (43.99)	240 (58.54)	93 (54.71)	35 (45.45)	
Age (months), median (IQR)	4 (2–8.5)	4 (2–9)	3 (2–8)	3.4 (1.6–6)	0.18
Co-morbidities, *n*(%) *n* (%)	94 (14.31)	56 (13.66)	28 (16.47)	10 (12.99)	0.64
Gestazional age, *n* (%)					
• <34 weeks	18/636 (2.83)	8/391 (2.05)	8/168 (4.76)	2/77 (2.60)	0.24
• 34–36 weeks	42/636 (6.60)	25/391 (6.39)	9/168 (5.36)	8/77 (10.39)	
• ≥37 weeks	576/636 (90.57)	358/391 (91.56)	151/168 (89.88)	67/77 (87.01)	
Chronic lung disease of prematurity, *n* (%)	7 (1.01)	2 (0.49)	5 (2.94)	0	0.02
Congenital heart disease, *n* (%)	16 (2.44)	13 (3.17)	2 (1.18)	1 (1.30)	0.42
Neuromuscolar Disease, *n* (%)	6 (0.91)	2 (0.49	4 (2.35)	0	0.07
Other comorbidities, *n* (%)	34 (5.18)	20 (4.88)	13 (7.65)	1 (1.30)	0.1
Siblings *n* (%)	292(44.44)	147(35.85)	98 (57.65)	47 (61.04)	<0.001
Palivizumab	3 (0.46)	3 (0.73)	0	0	0.67
Admitted to hospital, *n* (%)	368 (56.01)	231 (56.34)	71 (41.76)	66 (85.71)	<0.001
Admitted to PICU, *n* (%)	26 (3.96)	21 (5.12)	3 (1.76)	2 (2.60)	0.15
Admitted to PICU within the first 7 days since initial evaluation, *n* (%)	39 (5.94)	24 (5.85)	13 (7.65)	2 (2.60)	0.31
Need for nasogastric fluids *, *n* (%)	4 (0.61)	1 (0.24)	3 (1.76)	0	0.12
Intravenous fluids *, *n* (%)	221 (33.64)	105 (25.61)	59 (34.71)	57 (74.03)	<0.001
Need for Oxygen Low Flow *, *n* (%)	167 (25.42)	119 (29.02)	47 (27.65)	1 (1.30)	<0.001
Need for High Flow Oxygen *, *n* (%)	128 (19.48)	45 (10.98)	39 (22.94)	44 (57.14)	<0.001
CPAP *, *n* (%)	43 (6.54)	31 (7.56)	12 (7.06)	0	0.05
Mechanical ventilation *, *n* (%)	2 (0.30)	1 (0.24)	1 (0.59)	0	0.61
Salbutamol nebulization, *n* (%)	262 (39.88)	175 (42.68)	34 (20.00)	53 (68.83)	<0.001
Other bronchodilator, *n* (%)	55 (8.37)	52 (12.68)	3 (1.76)	0	<0.001
Corticosteroid treatment, *n* (%)	118 (17.96)	80 (19.51)	4 (2.35)	34 (44.16)	<0.001
Antibiotic treatment, *n* (%)	148 (22.53)	87 (21.22)	17 (10)	44 (57.14)	<0.001

***** Calculated if anytime needed within the first seven days since the initial hospital evaluation.

**Table 2 children-09-00491-t002:** Viral etiologies detection of at least one virus, on a total 264 children that underwent the test. RSV, respiratory syncytial virus.

Detection of Virus on RT-PCR (Based on 264 Children Tested with PCR, of Which 216 Had at Least One Positive Virus) *	Study Population	North *n* = 110	Middle *n* = 83	South *n* = 23	*p*-Value
RSV, *n* (%)	162 (75)	85 (77.27)	58 (69.88)	19 (82.61)	0.4
Rhinovirus, *n* (%)	48 (22.22)	8 (8.18)	34 (40.96)	5 (21.74)	<0.001
SARS-CoV-2 *, *n* (%)	18 (8.33)	16 (14.55)	2 (2.41)	0	0.003
Human-Metapneumovirus, *n* (%))	12 (5.56)	3 (2.73)	7 (8.43)	2 (8.70)	0.122
Parainfluenza, *n* (%)	8 (4.17)	6 (5.45)	2 (2.41)	1 (4.35)	0.61
Adenovirus, *n* (%)	2 (0.93)	0	2 (2.41)	0	0.35
Other viruses, *n* (%)	14 (6.48)	4 (3.64)	7 (7.23)	4 (17.39)	0.05
Coinfection, *n* (%)	43 (19.91)	13 (11.81)	25 (30.12)	5 (21.74)	0.006

* SARS-CoV-2 rapid test was carried out in all patients during first evaluation, while nasopharyngeal PCR test for SARS-CoV-2 was carried out in all admitted children.

**Table 3 children-09-00491-t003:** Disease severity according to main etiological group. CPAP, continuous positive airway pressure; HFNC, high-flow nasal cannulae; PICU, pediatric intensive care unit.

	RSV	Not-RSV	*p*-Value
	*n* = 162	*n* = 54 ^#^	
Sex, *n* (%)			0.43
• Female	76 (46.91)	22 (40.74)	
• Male	86 (53.09)	32 (59.26)	
Age (months), median (IQR)	2 (1–5)	4 (1.69–11)	0.004
Co-morbidities, *n* (%) *n* (%)	27 (16.67)	14 (25.93)	0.13
Gestazional age, *n* (%)			0.77
• <34 weeks	6/161 (3.73)	2/53 (3.77)	
• 34–36 weeks	13/161 (8.07)	6/53 (11.32)	
• ≥37 weeks	142/161 (88.20)	45/53 (84.91)	
Chronic lung disease of prematurity, *n* (%)	1 (0.62)	2 (3.70)	0.09
Congenital heart disease, *n* (%)	5 (3.09)	1 (1.85)	0.53
Neuromuscolar Disease, *n* (%)	3 (1.85)	2 (3.70)	0.68
Other comorbidities, *n* (%)	10 (6.17)	2 (3.70)	0.38
Siblings *n* (%)	96 (59.26)	27 (50)	0.23
Palivizumab	0	0	
Admitted to hospital, *n* (%)	130 (80.25)	37 (68.52)	0.06
Admitted to PICU, *n* (%)	9 (5.56)	3 (5.56)	0.61
Admitted to PICU within the first 7 days since initial evaluation, *n* (%)	22 (13.58)	4 (7.41)	0.7
Need for nasogastric fluids *, *n* (%)	3 (1.85)	0	0.42
Intravenous fluids *, *n* (%)	95 (58.64)	24 (44.44)	0.07
Need for Oxygen Low Flow, *n* (%)	74 (45.68)	18 (33.33)	0.11
Need for High Flow Oxygen *, *n* (%)	57 (35.19)	21 (38.89)	0.62
CPAP *, *n* (%)	24 (14.81)	3 (5.56)	0.05
Mechanica ventilation *, *n* (%)	1 (0.62)	1 (1.85)	0.43
Salbutamol nebulization, *n* (%)	65 (40.12)	15 (27.78)	0.1
Other bronchodilator, *n*(%)	14 (8.64)	2 (3.70)	0.19
Corticosteroid treatment, *n* (%)	38 (23.46)	7 (12.96)	0.1
Antibiotic treatment, *n* (%)	51 (31.48)	9 (16.67)	0.03

* Calculated if anytime needed within the first seven days since the initial hospital evaluation. # calculated on the number of single children with a non-RSV microbiological diagnosis (therefore, a total of 54 children with an alternative microbiological diagnosis, some of which had co-infections). Other bronchodilator, ipratropium; corticosteroid treatment, any intravenous or oral steroid treatment administered.

**Table 4 children-09-00491-t004:** Disease severity according to bronchiolitis sustained by a single virus or a co-infection. CPAP, continuous positive airway pressure; HFNC, high-flow nasal cannulae; PICU, pediatric intensive care unit.

	Single Agent	Coinfections	*p*-Value
	*n* = 173	*n* = 43	
Sex, *n* (%)			0.87
• Female	20 (46.51)	78 (45.09)	
• Male	23 (53.49)	95 (54.91)	
Age (months), median (IQR)	3 (1–7)	3 (1–6)	0.58
Co-morbidities, *n*(%) *n* (%)	32 (18.50)	9 (20.93)	0.72
• <34 weeks	4/171 (2.34)	4 (9.30)	0.11
• 34–36	16/171 (9.36)	3 (6.98)	
• ≥37	151/171 (88.30)	36 (83.72)	
Chronic lung disease of prematurity, *n* (%)	1 (0.58)	2 (4.65)	0.1
Congenital heart disease, *n* (%)	4 (2.31)	2 (4.65)	0.34
Neuromuscolar Disease, *n* (%)	4 (2.31)	1 (2.33)	0.67
Other comorbidities, *n* (%)	9 (5.20)	3 (6.98)	0.44
Siblings *n* (%)	94 (54.34)	29 (67.44)	0.12
Palivizumab	0	0	
Admitted to hospital, *n* (%)	137 (79.19)	30 (69.77)	0.18
Admitted to PICU, *n* (%)	9 (5.20)	3 (6.98)	0.44
Admitted to PICU within the first 7 days since initial evaluation, *n* (%)	19 (10.98)	7 (16.28)	0.34
Need for nasogastric fluids *, *n* (%)	3 (16.28)	0	0.51
Intravenous fluids *, *n* (%)	95 (54.91)	24 (55.81)	0.91
Need for Oxygen Low Flow *, *n* (%)	72 (41.62)	20 (46.51)	0.56
Need for High Flow Oxygen *, *n* (%)	60 (34.68)	18 (41.86)	0.38
CPAP *, *n* (%)	20 (11.56)	7 (16.28)	0.4
Mechanical ventilation *, *n* (%)	1 (0.58)	1 (2.33)	0.36
Salbutamol nebulization, *n* (%)	67 (38.73)	13 (30.23)	0.3
Other bronchodilator, *n* (%)	16 (9.25)	0	0.025
Corticosteroid treatment, *n* (%)	34 (19.65)	11 (25.58)	0.39
Antibiotic treatment, *n* (%)	44 (25.43)	16 (37.21)	0.12

* Calculated if anytime needed within the first seven days since the initial hospital evaluation. Other bronchodilator: ipratropium; Corticosteroid treatment: any intravenous or oral steroid treatment administered.

**Table 5 children-09-00491-t005:** Disease severity according to age. CPAP, continuous positive airway pressure; HFNC, high-flow nasal cannulae; PICU, pediatric intensive care unit.

216 Patients with an Identified Virus	12 Months or Less *n* = 193	>12 Months *n* = 23	*p*-Value
RSV, *n* (%)	147 (76.17)	15 (65.22)	0.25
Rhinovirus, *n* (%)	38 (19.69)	10 (43.48)	0.009
SARS-CoV-2, *n* (%)	18 (9.33)	0	0.12
Human-Metapneumovirus, *n* (%)	11 (5.70)	1(4.35)	0.79
Parainfluenza, *n* (%)	9 (4.66)	0	0.36
Adenovirus, *n* (%)	1 (0.50)	1 (4.35)	0.2
Other viruses, *n* (%)	8 (4.15)	6 (26.09)	0.001
Coinfection, *n* (%)	37 (19.17)	6 (26.9)	0.294
**657 total patients**	**12 months or less** ***n* = 548**	**>12 months** ***n* = 109**	***p*-Value**
Admitted to hospital, *n* (%)	315 (57.48)	53 (48.62)	0.09
Admitted to PICU, *n* (%)	21 (3.83)	5 (4.59)	0.72
Admitted to PICU within the first 7 days since initial evaluation, *n* (%)	34 (6.20)	5 (4.59)	0.34
Need for High Flow Oxygen *, *n* (%)	148 (27.01)	19 (17.43)	0.04
CPAP *, *n* (%)	35 (6.39)	8 (7.34)	0.71
Mechanical ventilation *, *n* (%)	1 (0.18)	1 (0.92)	0.3

* Calculated if anytime needed within the first seven days since the initial hospital evaluation.

**Table 6 children-09-00491-t006:** Disease severity according to gestational age. CPAP, continuous positive airway pressure; HFNC, high-flow nasal cannulae; PICU, pediatric intensive care unit.

214 CHILDREN with a Virus Isolated and Known Gestational Age	<34 Weeks *n* = 8	34–36 Weeks *n* = 19	37 or More *n* = 187	*p*-Value
RSV, *n* (%)	6 (75.00)	13 (68.42)	142 (75.94)	0.77
Rhinovirus, *n* (%)	2 (25.00)	5 (26.32)	40 (21.39)	0.65
SARS-CoV-2, *n* (%)	0	2 (10.53)	16 (8.56)	0.84
Human-Metapneumovirus, *n* (%)	1 (12.50)	0	11 (5.88)	0.39
Parainfluenza, *n* (%)	1 (12.50)	2 (10.53)	6 (3.21)	0.1
Adenovirus, *n* (%)	0	0	2 (1.07)	1
Other viruses, *n* (%)	2 (25.00)	0	12 (6.42)	0.08
Coinfection, *n* (%)	4 (50.00)	3 (15.79)	36 (19.46)	0.11
**636 children with known gestational age**	**<34 weeks** ***n* = 18**	**34–36 weeks** ***n* = 42**	**37 or more** ***n* = 576**	***p*-Value**
Admitted to hospital, *n* (%)	15 (83.33)	25 (59.52)	316 (54.86)	0.05
Admitted to PICU, *n* (%)	2 (11.11)	1 (2.38)	20 (3.47)	0.19
Admitted to PICU within the first 7 days since initial evaluation, *n* (%)	3 (16.67)	1 (2.38)	33 (5.73)	0.1
Need for High Flow Oxygen *, *n* (%)	7 (38.89)	13 (30.95)	105 (18.23)	0.015
CPAP *, *n* (%)	3 (16.67)	3 (7.14)	35 (6.08)	0.168
Mechanical ventilation *, *n* (%)	1 (5.56)	0	1 (0.17)	0.06

* Calculated if anytime needed within the first seven days since the initial hospital evaluation.

## Data Availability

Data available upon request to the corresponding author.

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
