# Peer review of "Epidemiology, Microbiology and Severity of Bronchiolitis in the First Post-Lockdown Cold Season in Three Different Geographical Areas in Italy: A Prospective, Observational Study"

_children, 2022, doi:10.3390/children9040491_

Round 1

Reviewer 1 Report

With great interest I have read this manuscript which reports important questions about bronchiolitis in Italy during the 2021-2022 cold season, when no lockdowns have been established. The paper is generally well-written, well-structured and clear. The authors should be congratulated for a very well-written paper. 

I have very few minor comments:

  1. Please provide keywords.
  2. Can you clarify co-infections group?

Author Response

Dear Editor and reviewers,

thank you very much for your efforts aimed at improving our paper.
We have provided below a point-by-point response to your comments, and all changes have been highlihghted in the revised version of the manuscript.

Reviewer 1

With great interest I have read this manuscript which reports important questions about bronchiolitis in Italy during the 2021-2022 cold season, when no lockdowns have been established. The paper is generally well-written, well-structured and clear. The authors should be congratulated for a very well-written paper. 

Thank you for appreciating our manuscript

I have very few minor comments:

  1. Please provide keywords.

Thank you, we have provided keywords

  1. Can you clarify co-infections group?

Thank you, we have now clarified it in the methods, as follow: “specifically, we defined as “co-infection group” those children that had more than one virus detected simultaneously at the nasopharyngeal swab” 

Reviewer 2 Report

In their manuscript: ”Epidemiology, microbiology and severity of bronchiolitis in the first post-lockdown cold season in three different geographical areas in Italy: a prospective, observational study” Camporeresi and colleagues investigate an important and very hot topic of epidemiological bronchiolitis trends during Covid pandemics.  The paper is very interesting and well written. There are some minor issues

  1. Material and methods: type of test used for the diagnosis of other viral etiologies should be mentioned as well as type of viruses that were investigated.
  2. For a non-Italian reader it would be helpful if demographic characteristic was performed for a whole group of patients, in addition to the division into separate centers
  3. Tables1, 3,4 – what “other bronchodilator” mean – this should be explained; corticosteroid treatment – should be explained systemic or inhaled.
  4. Discussion: “infecting these cells” – what cells exactly?
  5. English language proof reading is recommended; e.g. What does the sentence “In particular the four participating centers (page 4, verse 159)” mean? Two paragraps in the Discussion part 9page 13) should be rewritten.

Author Response

Dear Editor and reviewers,

thank you very much for your efforts aimed at improving our paper.
We have provided below a point-by-point response to your comments, and all changes have been highlihghted in the revised version of the manuscript.

Reviewer 2

In their manuscript: ”Epidemiology, microbiology and severity of bronchiolitis in the first post-lockdown cold season in three different geographical areas in Italy: a prospective, observational study” Camporeresi and colleagues investigate an important and very hot topic of epidemiological bronchiolitis trends during Covid pandemics.  The paper is very interesting and well written. There are some minor issues

Thank you for appreciating our manuscript

  1. Material and methods: type of test used for the diagnosis of other viral etiologies should be mentioned as well as type of viruses that were investigated.

Thank you, we have clarified it in the methods: “For non SARS-CoV-2 aetiology, we use the ePlex Respiratory Panel which detects PCR of RSV, Influenza, Rhinovirs, Adenovirus, Parainfluenza, Metapneumovirus, Bocavirus, Enterovirs

  1. For a non-Italian reader it would be helpful if demographic characteristic was performed for a whole group of patients, in addition to the division into separate centers

Thank you, we have now clarified that the first colomn of tables 1 and 2 are referred to the whole group of patients (study population)

  1. Tables1, 3,4 – what “other bronchodilator” mean – this should be explained; corticosteroid treatment – should be explained systemic or inhaled.

Thank you, we have clarified this in the appropriate tables: “Other bronchodilator: ipratropium; Corticosteroid treatment: any intravenous or oral steroid treatment administered.

  1. Discussion: “infecting these cells” – what cells exactly?

Thank you for spotting this mistake, we have now clarified that this is referred to upper respiratory tract epithelium

  1. English language proof reading is recommended; e.g. What does the sentence “In particular the four participating centers (page 4, verse 159)” mean? Two paragraps in the Discussion part 9page 13) should be rewritten.

Thank you, the paper has been english edited by Prof Roland, UK native speaker